# Immunogenicity and SARS-CoV-2 Infection following the Fourth BNT162b2 Booster Dose among Health Care Workers

**DOI:** 10.3390/vaccines11020283

**Published:** 2023-01-28

**Authors:** Yael Shachor-Meyouhas, Halima Dabaja-Younis, Avi Magid, Ronit Leiba, Moran Szwarcwort-Cohen, Ronit Almog, Michal Mekel, Avi Weissman, Gila Hyams, Vardit Gepstein, Nethanel A. Horowitz, Hagar Cohen Saban, Jalal Tarabeia, Michael Halberthal, Khetam Hussein

**Affiliations:** 1Pediatric Infectious Disease Unit, Rambam Health Care Campus, Haifa 3109601, Israel; 2The Ruth & Bruce Rappaport Faculty of Medicine, Technion–Israel Institute of Technology, Haifa 3200003, Israel; 3Management, Rambam Health Care Campus, Haifa 3109601, Israel; 4Department of Information Systems, The Max Stern Yezreel Valley College, Yezreel Valley 1930600, Israel; 5Epidemiology Unit, Rambam Health Care Campus, Haifa 3109601, Israel; 6Virology Laboratory, Rambam Health Care Campus, Haifa 3109601, Israel; 7Nursing Management, Rambam Health Care Campus, Haifa 3109601, Israel; 8Department of Pediatrics B, Rambam Health Care Campus, Haifa 3109601, Israel; 9Department of Hematology and Bone Marrow Transplantation, Rambam Health Care Campus, Haifa 3109601, Israel; 10Nursing Faculty, The Max Stern Yezreel Valley College, Yezreel Valley 1930600, Israel; 11Infection Control Unit, Rambam Health Care Campus, Haifa 3109601, Israel

**Keywords:** COVID-19, Immunogenicity, serology, health care workers, booster

## Abstract

Introduction: The fourth SARS-CoV-2 vaccine dose was found to protect against infection and more importantly against severe disease and death. It was also shown that the risk of symptomatic or severe disease was related to the antibody levels after vaccination or infection, with lower protection against the BA.4 BA.5 Omicron variants. The aim of our study was to assess the impact of the fourth dose on infection and perception of illness seriousness among healthcare workers (HCWs) at a tertiary health care campus in Haifa, Israel, and to investigate the possible protective effect of antibody levels against infection. Methods: We conducted a prospective cohort study among fully vaccinated HCWs and retired employees at Rambam Healthcare Campus (RHCC), a tertiary hospital in northern Israel. Participants underwent serial serological tests at 1, 3, 6, 9, 12 and 18 months following the second BNT162b2 vaccine dose. Only a part of the participants chose to receive the fourth vaccine. A multivariable logistic regression was conducted to test the adjusted association between vaccination, and the risk of infection with SARS-CoV-2. Kaplan–Meier SARS-CoV-2 free “survival” analysis was conducted to compare the waning effect of the first and second, third and fourth vaccines. Receiver Operating Characteristic (ROC) curve was plotted for different values of the sixth serology to identify workers at risk for disease. Results: Disease occurrence was more frequent among females, people age 40-50 years old and those with background chronic lung disease. The fourth vaccine was found to have better protection against infection, compared to the third vaccine; however, it also had a faster waning immunity compared to the third vaccine dose. Antibody titer of 955 AU/mL was found as a cutoff protecting from infection. Conclusions: We found that the fourth vaccine dose had a protective effect, but shorter than the third vaccine dose. Cutoff point of 955 AU/mL was recognized for protection from illness. The decision to vaccinate the population with a booster dose should consider other factors, including the spread of disease at the point, chronic comorbidities and age, especially during shortage of vaccine supply.

## 1. Introduction

For nearly 3 years COVID-19 has been challenging healthcare facilities and the whole population around the world. In the past year, Omicron has been circulating in the world with different variants and evidence for weakened and waning immunity among people vaccinated with the known available vaccines [1,2,3,4]. Recently, a vaccine containing Omicron BA.4 and BA.5 variants was approved by FDA and released for use [5]. Evidence suggests that the fourth vaccine dose provides higher protection from disease and particularly from severe disease and death compared with those who received only two or three vaccine doses [6,7,8,9]. Moreover, previous studies showed that the time since the last vaccine dose also contributes to the immunity to disease [6,8]. Finally, levels of antibodies after vaccination or infection were found to be negatively related to the risk of symptomatic or severe disease, with lower protection against the BA.4 BA.5 Omicron variants [10,11,12]. It was also shown by Barda et al. that antibody levels, even before the booster doses were administered, could be used as a prediction tool for estimating the risk of being infected [13].

Our study aimed to assess the humoral response to the fourth vaccine dose (second booster) and its effect on infection and perception of illness seriousness among HCWs at a tertiary health care center.

## 2. Methods

We conducted a prospective cohort study among fully vaccinated healthcare workers (HCW) and retired employees at Rambam Healthcare Campus (RHCC), a 1000-bed university hospital serving more than two million residents of Northern Israel and the only tertiary hospital in the region. RHCC has 5520 employees: 1220 physicians, 1880 nurses, 1137 paramedical workers and 1283 administrative workers.

The BNT162b2 vaccine was introduced in January 2021 to all HCWs in Israel including at RHCC. Study participants were recruited in two steps. First, all HCWs who were vaccinated with at least two BNT162b2 vaccine doses and had no history of known infection prior to the second vaccine dose administration were invited to participate in the study. Those who consented, underwent serial serological tests at 1, 3, 6, 9, 12 and 18 months after receiving the second vaccination dose (during February, April, July and October 2021, January and June 2022, respectively). A second recruitment time point was in January 2022, where all other HCWs were offered the opportunity to participate in the study, including those who were infected with SARS-CoV-2 before the vaccine was approved.

All participants completed a computerized questionnaire that included questions regarding some demographic characteristics, comorbidities, medications, allergic reaction or rash following vaccination, confirmed SARS-CoV-2 infection, or flu-like illness at each testing time point. In addition, all HCWs were guided to subjectively estimate the seriousness of their illness (based on personal experience), compared to other viral infection (flu-like) that they had in the past. The subjective seriousness score was categorized into mild, moderate and serious.

During December 2021 a fourth vaccine dose was offered to individuals for whom 4 months had passed since they received the third vaccine dose (Figure 1).

As a result, there were two distinct groups at the fourth and fifth serological testing time points at 9 and 12 months after the second dose: those who received three doses and those who received four doses (with or without previous SARS-CoV-2 infection).

The study was approved by the hospital’s Internal Review Board (#021-021), and a written informed consent was obtained from all participants.

### 2.1. Serology Assays

Serology testing was performed at 1, 3, 6, 9, 12 and 18 months post second vaccine dose on LIAISON^®^ XL analyzer with the LIAISON SARS-CoV-2 TrimericS IgG assay (DiaSorin S.p.A., Saluggia, Italy), according to the manufacturer’s instructions. This chemiluminescent immunoassay uses magnetic particles coated with recombinant trimeric SARS-CoV-2 spike protein for the quantitative determination of IgG antibodies. Cut-off values for positive serology were 22 AU/mL (Arbitrary Units/mL), boarder line 13–22 AU/mL; negative serology was reported for values <13 AU/mL. When needed (values >799 AU/mL), serum was diluted on-board 1:20 with LIAISON TrimericS IgG diluent.

The serology tests at the 6th point were performed between 20 June 20 to 22 June 2022.

### 2.2. Statistical Analysis

Descriptive statistics including mean, standard deviation, median, percentiles, counts and percentages were calculated for all the study variables. Normality tests for all the study quantitative variables were conducted using Shapiro–Wilk test. Whenever needed, parametric and non-parametric tests including one-way ANOVA, *t*-test and Kruskal–Wallis test were conducted according to the normality tests’ results. Differences between categorical variables were examined using the Fisher exact test or Pearson χ2 test. A multivariable logistic regression analysis was performed to test the adjusted association between the main independent variable, defined as level of vaccination (three categories: vaccine 1 and 2, vaccine 3 and vaccine 4), and the risk of infection with SARS-CoV-2. The association was tested with adjustment for sex, age, smoking, having a chronic lung disease and serology level.

Receiver operating characteristic (ROC) curve (with Youden index) was plotted to describe the relationship between the sensitivity and the false positive rate for different values of the sixth serology identifying workers at risk for disease.

Kaplan–Meier analysis was used to estimate SARS-CoV-2 free “survival” curves to health care workers who were vaccinated with two vaccines, three vaccines and four vaccines, where the outcome variable was time to infection with SARS-CoV-2.

For each result, *p* value < 0.05 was considered statistically significant. SPSS version 28 was used for all statistical analyses.

## 3. Results

### 3.1. Description of the Study Population

A total of 1696 HCWs were recruited to the study during February 2021, one month after receiving the second BNT162B2 vaccine dose. A serology test was performed at baseline. Additional serology tests were performed at 3, 6, 9, 12 and 18 months after the second vaccine dose in 1113, 1058, 986, 939 and 697 participants, respectively. (Figure 1). A total of 461 HCWs attended all six examination points.

During the second enrollment time point in January 2022, 331 HCWs signed an informed consent and joined the two last serology tests (5th and 6th) at 12 months and 18 months, respectively.

### 3.2. SARS-CoV-2 Infection among Participants

During the study period, a total of 323 HCWs were tested positive for SARS-CoV-2, of whom 5 (1%) were positive before the vaccine was available, 4 (1%) were positive after receiving the first dose, 26 (8% of the positive cases) after receiving the second dose, 241 (75%) after receiving the third dose and 47 (15%) after receiving the fourth dose.

In a follow-up performed six-month after the third and fourth vaccine doses, 134 HCWs out of those who received the third vaccine dose (28.3%) were found to be positive for SARS-CoV-2 after the third dose, and 47 HCWs out of who received the fourth vaccine dose (25%) were found to be positive for SARS-CoV-2 after the fourth dose.

### 3.3. Multivariate Analysis of the Independent Association between the Number of Vaccine Doses and the Risk of SARS-CoV-2

Adjusting for sex, age, smoking, underlying chronic lung disease and antibody level, the number of vaccine doses had a significant association with the risk of being infected with SARS-CoV-2 in a dose response manner, comparing vaccine three and vaccine four to one and two vaccines (Table 1). In addition, female sex, age group 40–50 years and underlying chronic lung disease were associated with higher risk of becoming infected. No association was found between smoking and the risk of infection with SARS-CoV-2 (Table 1).

When comparing the HCW who were negative for SARS CoV-2 to those who were positive within the first 6 months after their last vaccine dose and those who were positive after 6 months from their last vaccine dose (Table 2), there was a significant difference in the age which was higher among those who were negative for SARS CoV-2. Women also tended to be more positive for SARS-CoV-2 than men especially in the first 6 months after vaccination and those with chronic lung disease were less likely. The antibody levels were also much lower among those who were not sick.

### 3.4. ROC Analysis

ROC analysis using the Youden index was applied with a cutoff of 955 AU/mL as the critical point for infection, i.e., a serology result of 955 AU/mL and above was associated with better protection against infection. Area under the curve was 0.73, standard error was 0.019 and 95% CI 0.692–0.767; *p* < 0.001 (Figure 2).

In addition, the average serology levels among HCW who were positive for SARS-CoV-2 (less than six month after vaccination and more than six months after vaccination) were much higher than those who weren’t ill (2410 and 2220 vs. 702, respectively, *p* < 0.001) (Table 2).

### 3.5. Kaplan–MeierSARS-CoV-2 Free “Survival” Analysis

Kaplan–Meier SARS-CoV-2 free “survival” analysis (Figure 3) demonstrates that workers vaccinated with the third vaccine dose had better SARS-CoV-2 free survival (i.e., not being infected) compared to workers vaccinated with two vaccines (the first and the second vaccine doses). However, workers vaccinated with the fourth vaccine dose had the lowest SARS-CoV-2 free “survival” time compared to those who vaccinated with the third vaccine and the one and two vaccines.

### 3.6. Perception of Severity of Illness

None of the study participants were hospitalized due to COVID-19. All participants whose tests for SARS-CoV-2 were positive were asked to describe their perceived illness seriousness according to their subjective and personal experience of illness, in comparison with their past flu-like illness experiences. Of the 300 that rated their illness (93% of those who were positive for SARS-CoV-2), there was no significant difference in the proportion of mild illness perceived by those who had received two, three and four doses of vaccine (52%, 60% and 66%, respectively, *p* = 0.49). There was also no correlation between the seriousness of COVID-19 and the level of SARS-COV-2 antibodies up to 3 months before infection and even to the serology available within the 30 days before infection.

### 3.7. Serology Levels

A total of 697 HCW had their 6th SARS-COV-2 serology test in June 2022. Their characteristics are presented in Table 3.

The mean SARS-COV-2 antibody levels at 6th serology test point was 2694.8 (range 6–24,500), SD 3369.8.

A weak linear correlation between age and serology level was observed, where older age was associated with higher serology level (*p* = 0.001, r = 0.127), but after dividing the age variable into age group categories, no statistically significant difference between age groups was observed.

BMI was recorded for 555 (80%) participants. No association was found between serology level and BMI.

There were 140 HCWs who were not infected with SARS-COV-2 and received the fourth vaccine. Among these HCWs, a negative association was found between smoking and median serology level, where smokers had a lower median serology levels compared to non-smokers (IQR = 1629, Median = 517 vs. IQR = 2964, Median = 1060; *p* = 0.027). No associations were found between age, sex and serology level among these HCWs.

There were 244 HCW among those who attended all 6 serology tests and were not positive for SARS-COV-2 by June 2022. Their serology tests are presented in Figure 4 (arrows describe the third and fourth vaccine dose that were offered).

## 4. Discussion

In this study we aimed to assess the humoral response to the fourth vaccine dose and its impact on infection and perception of illness among our health care workers.

Our results show that the average age of those who were ill was lower than the average age of those who were not ill. Possible explanations for this finding are that many of the older HCW received the fourth vaccine dose, and therefore were more protected, and that older HCWs were more careful during the pandemic compared to younger HCWs. These explanations are supported by other studies, showing that older people were more likely to comply with vaccinations compared to younger people [9], and perhaps more likely to comply with measures such as isolation and social distancing [14]. The same explanations can be applied to those with chronic lung disease, that may be more likely to get vaccinated and to comply with recommended measures [15]. Those with chronic lung disease were positive for SARS-CoV-2 after more than 6 months after vaccination, a fact that may be explained by waning immunity in this group or by more SARS-CoV-2 tests taken by individuals at-risk. Waning immunity has been demonstrated in the literature in both the third vaccine dose and the fourth vaccine dose [2,13,16].

Our results demonstrated that females were more prone to disease than men, especially in the first 6 months after vaccination. This finding can partly be explained by the fact that most women among our cohort are at the childbearing age, and usually take care of their children and stay with them while they are sick, as was found by another study among our HCWs, which was conducted before vaccines were available [17]. Another possible explanation is that women may take more tests compared to men, to ensure that they are not ill. In addition, women may be less likely to be vaccinated with the fourth vaccine dose, especially those in the birth giving ages. This explanation is supported by an Israeli study performed by Cohen et al., testing the association between receiving a fourth vaccine dose and SARS-CoV-2 infection, where it was demonstrated that men were more likely to take the fourth vaccine dose than women [9]. This explanation is also supported by our results, demonstrating a statistically significant difference in the rate of women vaccinated with the fourth vaccine compared to men (23% vs. 39%, respectively, *p* < 0.0001). It is worth mentioning that there are known sex differences with regard to disease severity, such that males are more prone to severe disease, and females are more prone to long Covid-19 [18]. This may also explain the sex differences in compliance to vaccine. We also demonstrated in the past sex differences in antibody levels, while women had higher levels [19]. In our study, no sex differences were found with regard to antibody levels after the fourth vaccine among non-infected participants. Sex differences regarding susceptibility to disease should be further evaluated.

No differences were found in the perceived seriousness of illness between those who were vaccinated with the third or fourth vaccine dose in the first 6 months, implying the importance of the third vaccine dose, although there are studies that demonstrating the importance of the fourth vaccine dose especially during the first period and while there is a peak in the virus spread [6,7,8,9]. It is particularly important for the elderly and those with chronic illnesses. In times of shortage of vaccines, the priority for booster vaccines should be the elderly, chronically ill people and susceptible patients, as demonstrated by some studies, which showed that the booster protected from hospitalization and death in Israel and in other countries [4,6,8].

The third vaccine dose was also followed by a rapid rise in antibody titer, as was demonstrated by our results and by other studies [20,21]. However, as shown in our results, the rise after the fourth dose is milder, and, therefore, the differences between third and fourth dose should be further studied, especially in cases of shortage in supply or countries with poor resources. Barda et al., have found a decline in antibody level and a higher risk for infection four months after receiving the fourth vaccine [13]. They also found that a cutoff value of 700 BAU in the antibody level had a protective effect, and was associated with 35% reduced infection rate after six months [13]. In our study, we have found that a cutoff value of 955 AU/mL had a protective effect from disease. This value is much higher than the maximal value of the original test kit, before it was diluted (799 AU/mL).

The use of serology test to prioritize or tailoring a vaccination strategy is complex and laboring. The low serology levels in patients with comorbidities may be helpful to determine the need for a booster, but this should be further studied. The correlation between the antibody levels and protection was demonstrated in several studies and was already established [10,12,13].

Our results demonstrate a better protection of the fourth vaccine against being infected, compared to the third vaccine, as reflected by the multiple logistic regression. Nevertheless, we demonstrated a faster waning immunity of the fourth vaccine compared to the third vaccine, as reflected by the Kaplan–Meier SARS-CoV-2 free “survival” curves. It may be explained by the fact that the fourth vaccine dose was introduced at the beginning of the Omicron variant outspread and had a protective effect against old variants which were still circulating. With the prompt outspread of the Omicron variant, the protection of the fourth vaccine, which was not efficient against infection with the Omicron variant decreased [1,2]. This reduction in protection in comparison with the third vaccine dose was also observed in a study conducted by Cohen et al. studying almost 30,000 HCWs in 11 hospitals in Israel [9], and by other studies [13].

With regard to the seriousness of disease, our HCWs cohort are mostly contains healthy and young people. No member of the cohort was hospitalized during the study period. Yet, the subjective feeling and the perception of disease seriousness did not differ between those who received three and four doses. This observation is unique, since it shows that in a healthy population, although there is a decrease in the booster’s ability to protect from the disease, the perception of seriousness may not change.

Another interesting observation is the fact that among some of our HCWs who did not have known infection with SARS-CoV-2, the antibody levels were extremely high (Figure 4), suggesting the possibility of asymptomatic infection that was un-noticed especially among fully vaccinated people. Asymptomatic infection is well known and documented [22,23], and the need for routine tests (PCR or antigen) during a rise in SARS-CoV-2 cases or other situation during the pandemic should be carefully considered, while taking into account their pros and cons [22,23,24]. Extremely high serology level among individual who was not been vaccinated lately can raise the suspicion of asymptomatic infection, and therefore establishing a cutoff for disease may exclude the need for another booster. However, the use of serological tests for this purpose for the entire population is not justified.

Our study has several limitations. First it is a single center study reflecting the population of our HCWs. Second, in our cohort, although consisted of adults and older volunteers, most health care workers were younger than 60 years old and were usually healthier than the general population. In addition, most of our cohort members were women, as our HCWs population consists more women. Third, the antibodies examined are not neutralizing antibodies although a relationship has already been demonstrated [25]. Fourth, the HCWs who were negative for SARS-CoV-2 might have asymptomatic infection in addition to the vaccination although each HCW was tested upon clinical symptoms and after contact with positive patients or relatives.

## 5. Conclusions

Our study demonstrated a protective effect of the fourth vaccine dose but shorter compared with the third among HCWs. There was no difference in the seriousness of the illness perceived by the HCWs. Women were more prone to SARS-CoV-2 infection, as well as younger HCWs and HCWs with chronic lung disease. A cutoff point of 955 AU/mL in antibody level was demonstrated as protective against illness. The decision whether to vaccinate the population with a booster dose at a specific point may take into consideration other factors, including the spread of disease at this point, background chronic diseases, age and circulating variants.

## Figures and Tables

**Figure 1 vaccines-11-00283-f001:**
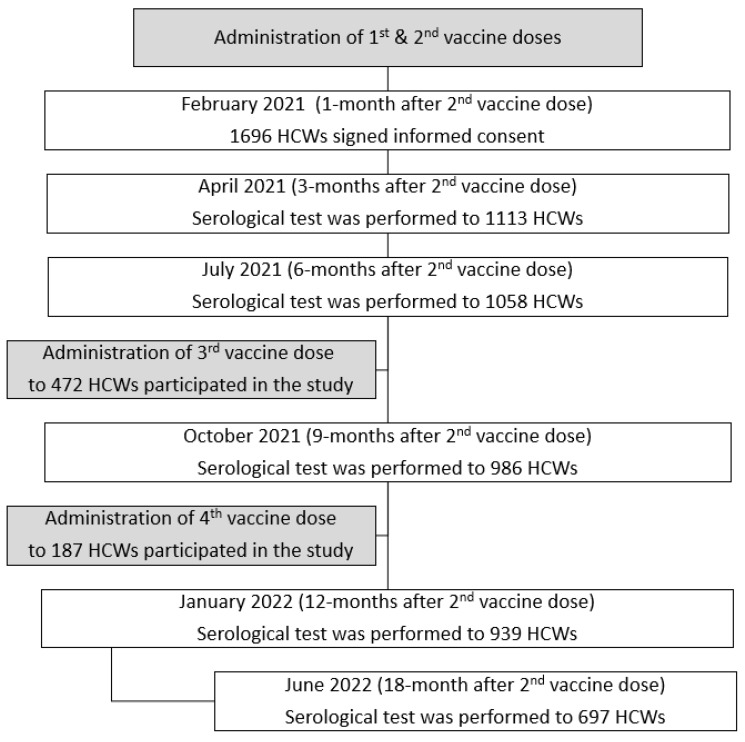
The serology cohort recruitment examination tests. A total of 461 Health Care Workers attended all 6 serology tests.

**Figure 2 vaccines-11-00283-f002:**
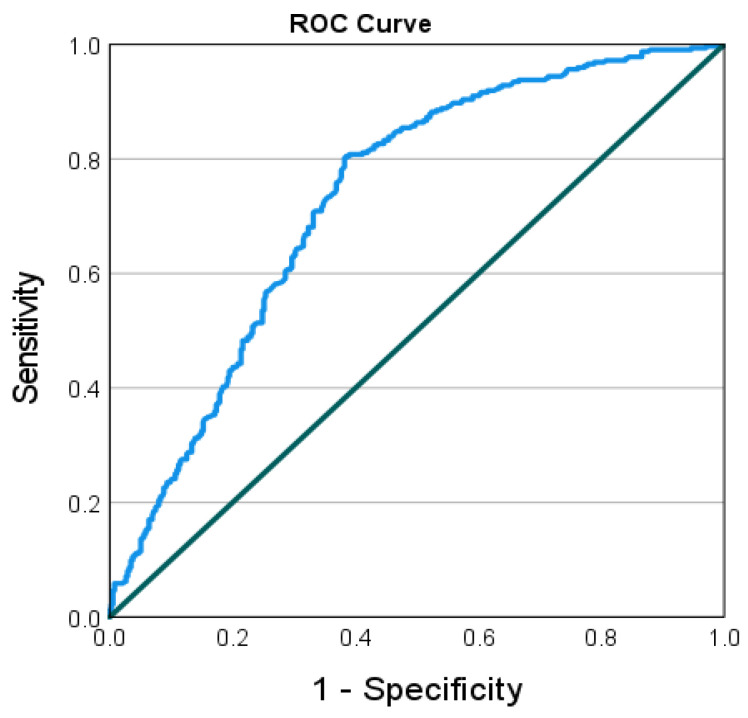
ROC curve (with Youden index) describing the relationship between the sensitivity and the false positive rate for different values of the sixth serology identifying workers at risk for disease.

**Figure 3 vaccines-11-00283-f003:**
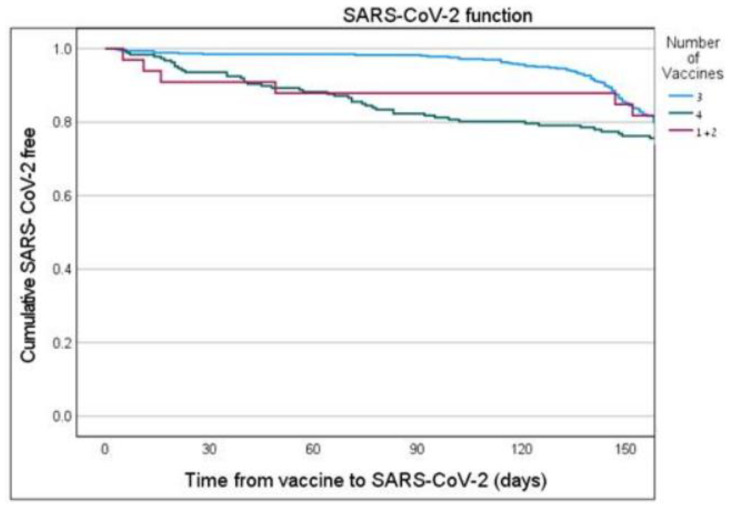
Disease Free Survival analysis of HCW after vaccine. *p* = 0.017 between 1+2 vaccine vs. 3 vaccines.

**Figure 4 vaccines-11-00283-f004:**
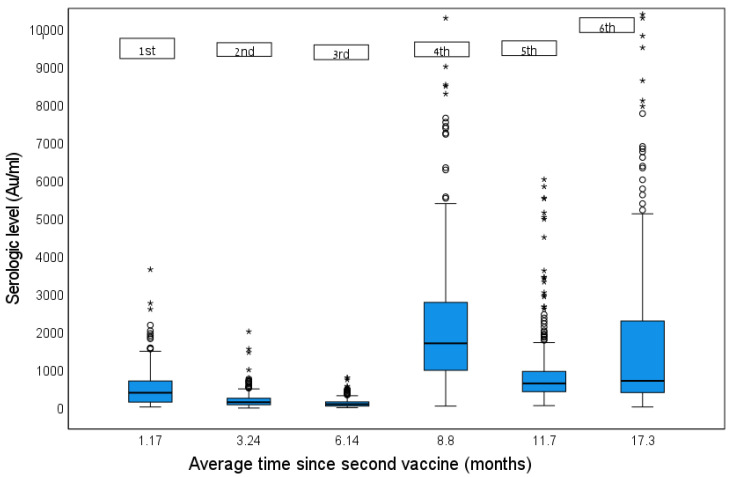
Serial serological tests among HCW who were negative for COVID-19 during the study period. Straight line represents the 3rd vaccine (Booster 1); Dashed line represents the 4th vaccine (Booster 2); * represents extreme outliers (data points); o represents mild outliers (data points).

**Table 1 vaccines-11-00283-t001:** Multivariate analysis of the independent factors associated with increased risk of SARS-CoV-2 infection.

Variables	OR	95% CI	*p*
Sex			
Male	1		
Female	1.99	1.30–3.05	0.002
Age (years)			
60+	1		
50–60	1.30	0.80–2.10	NS
40–50	1.77	1.07–2.92	0.026
<40	1.58	0.89–2.81	NS
Smoking			
Smoker	1		
Non-smoker	0.62	0.34–1.14	NS
Chronic lung disease			
No	1		
Yes	2.99	1.05–8.56	0.041
Number of vaccines			
Vaccine 1 and 2	1		
Vaccine 3	0.18	0.08–0.41	<0.0001
Vaccine 4	0.05	0.02–0.12	<0.0001
Serology #6	1 *	1	<0.0001

* Do not contribute to the model.

**Table 2 vaccines-11-00283-t002:** Comparison between HCWs who were not infected with SARS-CoV-2, HCWs who became infected within 6 months of the last vaccine dose, and HCWs who became infected later than 6 months after the last vaccine dose.

	SARS-CoV-2 Negative (N = 374, 54.0%)	SARS CoV-2 Positive ≥ 6 m (N = 105, 15.0%)	SARS CoV-2 Positive < 6 m (N = 213, 31.0%)	P
Age (years)	52.9 ± 11.4	49.8 ± 10.03	50.5 ± 10.14	P^1^ = 0.032 P^2^ = 0.034 P^3^ = NS
Age group (years) <40 40–50 50–60 60+	53 (14%) 94 (25%) 123 (33%) 104 (28%)	18 (17%) 35 (34%) 38 (36%) 14 (13%)	34 (16%) 65 (30%) 65 (31%) 49 (23%)	P = 0.081
Sex Male Female	109 (29%) 265 (71%)	24 (23%) 81 (77%)	31 (15%) 182 (85%)	P < 0.001 P^1^ = NS P^2^ < 0.001 P^3^ = NS
BMI	26.0 ± 4.2	25.5 ± 4.7	26.04 ± 5.12	P = NS
Heart disease	21 (5.6%)	9 (8.6%)	12 (5.6%)	P = NS
Chronic lung disease	7 (1.9%)	10 (9.5%)	5 (2.3%)	P < 0.001 P^1,3^ = 0.008 P^2 =^ NS
Active oncologic disease	1 (0.3%)	1 (0.95%)	1 (0.47%)	NA
Autoimmune disease	27 (7.2%)	6 (5.7%)	17 (8.0%)	P = NS
Intestinal disease	5 (1.3%)	3 (2.9%)	3 (1.4%)	P = NS
Hypothyroidism	46 (12.3%)	6 (5.7%)	25 (11.7%)	P = NS
Chronic renal failure	2 (0.5%)	0 (0%)	1 (0.47%)	NA
Treatment with dialyzes	0 (0%)	0 (0%)	1 (0.47%)	NA
Smoking	42 (11.2%)	12 (11.4%)	12 (5.6%)	P = 0.066
Immunosuppressive therapy	8 (2.1%)	2 (1.9%)	5 (2.3%)	P = NS
*Sixth serology level	702 [385–2200]	2220 [988–5105]	2510 [1220–4755]	P^1,2^ < 0.001 P^3^ = NS

P^1^ = SARS-CoV-2 Negative vs. SARS-CoV-2 Positive ≥ 6 months, P^2^=SARS-CoV-2 Negative vs. SARS-CoV-2 Positive < 6 months; P^3^=SARS-CoV-2 Positive ≥ 6 months vs. SARS-CoV-2 Positive < 6 months. Five females were excluded from this table since they were ill before the first vaccine. * Serology level among positive HCW represents infection and vaccination influence and cannot be separated.

**Table 3 vaccines-11-00283-t003:** Characteristics of participants who went through the 6th serology test in June 2022.

N = 697	
Mean age in years ± sd (range)	51.7 ± 10.8 (19.7–86.6)
Age groups Till 30 30–40 40–50 50–60 60+	16 (2%) 89 (13%) 194 (28%) 229 (33%) 169 (24%)
Gender Male Female	164 (23.5%) 533 (76.5%)
Mean serology level ± sd (median)	2694.8 ± 3369.8 (1370)
Mean BMI ± sd (range)	25.9 ± 4.59 (17.2–49.0)
BMI categories ≤24.9 25–30 ≥30	252 (45.5%) 212 (38%) 91 (16.5%)
Heart disease	42 (6%)
Chronic lung disease	22 (3%)
Active oncologic disease	21 (3%)
Autoimmune disease	50 (7.2%)
Intestinal disease	11 (1.6%)
Hypothyroidism	77 (11.0%)
Chronic renal failure disease	3 (0.4%)
Treatment with dialyzes	1 (0.1%)
Smoking	67 (9.6%)
Receiving immunosuppressive treatments	15 (2.2%)

## Data Availability

The data are not publicly available due to privacy and ethical restrictions.

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
