# Peer review of "Immunogenicity and SARS-CoV-2 Infection following the Fourth BNT162b2 Booster Dose among Health Care Workers"

_vaccines, 2023, doi:10.3390/vaccines11020283_

Round 1

Reviewer 1 Report

Shachor-Meyouhas et al. report a study aimed at assessing the humoral response to the 4th BNT162b2 vaccination and its effects on infection and perception of illness among health care workers. Despite several study design limitations, correctly pointed out in the discussion, the article may still be of some interest. Nonetheless, result description may be improved here and there, for the sake of clarity.

Tables and Figures should be carefully revised. For instance, legend to graph in Fig. 2 reads “number of vaccines 3.00, 4.00, 12.00” whereas it should probably read “number of vaccines 3, 4, 1+2”. In table 2, only p values that are statistically significant should be in bold, and the meaning of bold should be explained in the foonote. English should be revised throughout, as there are quite several typos, wrong/interrupted sentences or missing words. In Fig. 3, labelling to the X axis would be more informative if months from the second immunization were shown, rather than test number; the number of participants tested at each time point should also be indicated here. AU should be defined. In the Discussion, lines 255-257 are not clear. When referring to Kaplan-Meier “survival curve” (for example at lines 162 and 266), it should be clearly stated that it is a SARS-CoV-2-free “survival curve”, as no individuals actually died during the study. And so on…

Author Response

Dear reviewer,

We would like to thank you for your remarks which helped us to improve our article.

Following is our reply.

Wherever line numbers are mentioned, we refer to the original line numbers mentioned by the reviewer (of the version reviewed by the reviewer).

Shachor-Meyouhas et al. report a study aimed at assessing the humoral response to the 4th BNT162b2 vaccination and its effects on infection and perception of illness among health care workers. Despite several study design limitations, correctly pointed out in the discussion, the article may still be of some interest. Nonetheless, result description may be improved here and there, for the sake of clarity.

Tables and Figures should be carefully revised. For instance, legend to graph in Fig. 2 reads “number of vaccines 3.00, 4.00, 12.00” whereas it should probably read “number of vaccines 3, 4, 1+2”. In table 2, only p values that are statistically significant should be in bold, and the meaning of bold should be explained in the footnote. English should be revised throughout, as there are quite several typos, wrong/interrupted sentences or missing words. In Fig. 3, labelling to the X axis would be more informative if months from the second immunization were shown, rather than test number; the number of participants tested at each time point should also be indicated here. AU should be defined. In the Discussion, lines 255-257 are not clear. When referring to Kaplan-Meier “survival curve” (for example at lines 162 and 266), it should be clearly stated that it is a SARS-CoV-2-free “survival curve”, as no individuals actually died during the study. And so on…

Tables and figures were corrected and are placed in the article.

All tables were corrected. P-value for non-significant results was marked as “NS”.

English was revised and corrected accordingly.

AU was defined in the methods section as Arbitrary Units.

Discussion-Lines 255-257 were rephrased to present the information in a clearer manner

When referring to Kaplan-Meier “survival curve” (for example at lines 162 and 266), it should be clearly stated that it is a SARS-CoV-2-free “survival curve”, as no individuals actually died during the study. And so on…

We have corrected the statement regarding the Kaplan-Meier wherever the term appears in the text.

Reviewer 2 Report

The following issues are suggested to be clarified:

1.     Tables are not in well ordered. Please fix it.

2.     How did the authors determine this statement? Line 71-72: “…with at least two BNT162b2 vaccine doses and had not been infected prior to the 2nd vaccine dose administration were invited to participate in the study.” Did the authors collect blood samples and tested for SARS-Cov-2 antigens or blood antibodies? this requires to be clearly stated.

3.     It would be interesting if the authors could analyze the differences in vaccine-induced antibody responses with age, and smoke. Were the vaccine-induced neutralizing antibody responses or anti-RND antibody levels observed in elderly individuals lower than those in younger individuals?

4.     In line237-238: “We also demonstrated in the past sex differences in antibody levels, while women had higher levels”. The author should discuss this with the comparison to current COVID-19 vaccines. 

Author Response

Dear reviewer,

We would like to thank you for your remarks which helped us to improve our article.

Following is our reply.

Wherever line numbers are mentioned, we refer to the original line numbers mentioned by the reviewer (of the version reviewed by the reviewer).

The following issues are suggested to be clarified:

  1. Tables are not in well ordered. Please fix it.

Thank you, we have fixed it, and tables are in the correct order now.

  1. How did the authors determine this statement? Line 71-72: “…with at least two BNT162b2 vaccine doses and had not been infected prior to the 2nd vaccine dose administration were invited to participate in the study.” Did the authors collect blood samples and tested for SARS-Cov-2 antigens or blood antibodies? this requires to be clearly stated.

Lines 71-72 were corrected to mention that HCWs that did not have a history of KNOWN infection were recruited. In addition, it should be pointed out that all HCWs were offered an antibody test before vaccines were introduced. More than 80% of HCWs were tested, and only 2% of them were found to be asymptomatically positives (without receiving the vaccine). Moreover, all employees who had symptoms, or contacted positive persons during the study period were tested (PCR test).

  1. It would be interesting if the authors could analyze the differences in vaccine-induced antibody responses with age, and smoke. Were the vaccine-induced neutralizing antibody responses or anti-RND antibody levels observed in elderly individuals lower than those in younger individuals?

It should be noted that the antibody response was examined with a different test than anti-RND antibody. However, we analyzed the differences in vaccine-induced antibody responses with age and smoking, while comparing the serology levels of those who were vaccinated with the 4th vaccine to the serology levels of those who were not vaccinated with the 4th vaccines (both groups did not have known infection history). We found a negative association between smoking and median serology level, where smokers had a lower median serology level compared to non-smokers. No associations were found between age, sex and serology level among these HCWs. These analyses were added to the “results” section and discussed.

  1. In line237-238: “We also demonstrated in the past sex differences in antibody levels, while women had higher levels”. The author should discuss this with the comparison to current COVID-19 vaccines.

Discussion about Sex differences in antibody levels with regard to the current COVID-19 vaccine was added to the manuscript text.

Reviewer 3 Report

In this paper the Authors analyze a large number of healthcare workers from a single hospital in Israel in terms of protection from infection, illness severity and level of protective antibodies after vaccination with the mRNA vaccine against SARS-CoV-2.

The paper adds to a series of analogous papers in the existing literature on this topic and does not add any really novelty in the field except for the protective antibody titer which would be recognized in 955 AU/ml by the application of the Youden–index (which is not reported as a figure by itself). It is curious that a similar approach was never applied in other researches of level and standardization of a protective antibody level still lacks. Thus, the Authors need to better specify their method and compare (and discuss) their results. Moreover, their paper is plenty of errors, p values in the tables are difficult to understand and sentences without ending are present.

Author Response

Dear reviewer,

We would like to thank you for your remarks which helped us to improve our article.

Following is our reply.

Wherever line numbers are mentioned, we refer to the original line numbers mentioned by the reviewer (of the version reviewed by the reviewer).

In this paper the Authors analyze a large number of healthcare workers from a single hospital in Israel in terms of protection from infection, illness severity and level of protective antibodies after vaccination with the mRNA vaccine against SARS-CoV-2.

The paper adds to a series of analogous papers in the existing literature on this topic and does not add any really novelty in the field except for the protective antibody titer which would be recognized in 955 AU/ml by the application of the Youden–index (which is not reported as a figure by itself). It is curious that a similar approach was never applied in other researches of level and standardization of a protective antibody level still lacks. Thus, the Authors need to better specify their method and compare (and discuss) their results. Moreover, their paper is plenty of errors, p values in the tables are difficult to understand and sentences without ending are present.

Thank you very much for your review and remarks.

We believe that the Cutoff point that was find is an important finding that should be further evaluated and studied. We have mentioned another study that pointed out a cutoff but there weren’t any others. We added the figure of the ROC curve (with Youden–index) to our manuscript.

Other mistakes and errors were corrected including corrections of the tables and placing the P value and NS instead of non -significant values in order to present the results in a clearer manner.

Reviewer 4 Report

Thank you to write an interesting topic of COVID-19 vaccine

Reviewer’s comments

The manuscript present data on immunogenicity and SARS-CoV-2 infection following the 4th BNT162b2 among HCW. It is important topic but unfortunately the manuscript is difficult to understand with some misunderstanding in the concept of assessing the disease severity by the participants. I also highlight ‘’the comparation of participants without and with COVID-19 during the study’. Why the authors didn’t exclude the infected participants since the COVID-19 infection will obscure the level of the antibody due to vaccination? Its widely accepted that COVID-19 infection increased the antibody level naturally, but the authors analyzed and compare with non-infected participants. How the authors assess which antibody level induced by vaccination among the infected participants?

Major consideration:

line 81. Methods section: no presentation of your methods in educate the participants to categorized their illness severity. In my opinion, educate the public in assessing any disease severity need a formal education (medical), please clarify

line 177. The comparison of antibody level among infected and non-infected participants in this study is risky. We couldn’t evaluate the impact of vaccination itself.

line 185. None of the participant hospitalized due to COVID-19. And the authors also asked the participants to describe their perceived illness severity as mild, moderate or severe. Two contradictive statements since the moderate to severe COVID-19 cases should be in-patient. Please clarify

Author Response

Dear reviewer,

We would like to thank you for your remarks which helped us to improve our article.

Following is our reply.

Wherever line numbers are mentioned, we refer to the original line numbers mentioned by the reviewer (of the version reviewed by the reviewer).

Reviewer’s comments

The manuscript present data on immunogenicity and SARS-CoV-2 infection following the 4th BNT162b2 among HCW. It is important topic but unfortunately the manuscript is difficult to understand with some misunderstanding in the concept of assessing the disease severity by the participants. I also highlight ‘’the compartion of participants without and with COVID-19 during the study’. Why the authors didn’t exclude the infected participants since the COVID-19 infection will obscure the level of the antibody due to vaccination? Its widely accepted that COVID-19 infection increased the antibody level naturally, but the authors analyzed and compare with non-infected participants. How the authors assess which antibody level induced by vaccination among the infected participants?

Thank you for your remarks. We compared those who were vaccinated and positive for SARS-CoV-2 to those who stayed negative in order to understand the risk factors for being positive for SARS-CoV-2. There is no doubt that disease will influence antibody levels but the comparison was for other factors and the antibody level that was found to be protective was mentioned in the text.

We added the following remark in table 2 to emphasize it:

“*Serology level among positive HCW represent infection and vaccination influence and cannot be separate”

The education of the HCW regarding how to reply to the seriousness perception is mentioned below, along with explanation in the method section.

In addition, the term “severe” was replaced by the term “serious” to emphasize to emphasis that it is a subjective variable based on participants’ perceptions.

Major consideration:

line 81. Methods section: no presentation of your methods in educate the participants to categorized their illness severity. In my opinion, educate the public in assessing any disease severity need a formal education (medical), please clarify

Thank you for the remark, it is important to clarify. 

All HCW were guided to subjectively (personally experience) their severity of illness in comparison with their other viral infection (flu-like) in the past. The severity subjective score was: mild, moderate and serious (we have changed the word in English in order to avoid misunderstanding since in Hebrew the words “severe” and “serious” are expressed by the same word). We have added the data in the method section along with explanation that it is a personally experience and not a medical scoring system. We believe it is important data (although self-reported) since there was no influence of the 4th vaccine dose on self-experience illness in comparison to other past flu-like illnesses. 

line 177. The comparison of antibody level among infected and non-infected participants in this study is risky. We couldn’t evaluate the impact of vaccination itself.

Thank you very much. We agree and added to table 3 an explanation regarding that.

*Serology level among positive HCW represent infection and vaccination influence and cannot be separate

line 185. None of the participant hospitalized due to COVID-19. And the authors also asked the participants to describe their perceived illness severity as mild, moderate or severe. Two contradictive statements since the moderate to severe COVID-19 cases should be in-patient. Please clarify

Thank you, we have clarified it also in the results section and also in the discussion part by stressing in bold the word subjective.

Round 2

Reviewer 3 Report

in the revised form the authors have satisfactorily answered 

Reviewer 4 Report

Thank you to revised the manuscript